# Assessing tobacco use in Swedish young adults from self-report and urinary cotinine: a validation study using the BAMSE birth cohort

Anna Zettergren ![ORCID],[1] Shanzina Sompa,[1] Lena Palmberg,[1] Petter Ljungman,[1,2] Göran Pershagen,[1] Niklas Andersson,[1] Christian Lindh,[3] Antonios Georgelis,[1,4] Inger Kull,[5,6] Erik Melen,[1,5,6] Sandra Ekström,[1,4] Anna Bergstrom[1,4]

For numbered affiliations see end of article.

**Correspondence to**
Anna Zettergren;
anna.zettergren@ki.se

## ABSTRACT

**Objectives** Studies on health effects of tobacco often rely on self-reported exposure data, which is subjective and can lead to misclassification. The aim of this study was to describe the prevalence of cigarette smoking, snus and e-cigarette use, as well as to validate self-reported tobacco use among young adults in Sweden.

**Method** Participants of a population-based Swedish cohort (n=3052), aged 22–25 years, assessed their tobacco use in a web questionnaire. Urinary cotinine was analysed in a subsample of the study population (n=998). The agreement between self-reported tobacco use and urinary cotinine was assessed using Cohen's Kappa coefficient (κ) at a cut-off level of 50 ng/mL.

**Results** Patterns of tobacco use differed between men and women. Among men, 20.0% reported daily snus use, 5.8% daily cigarette smoking and 5.6% any e-cigarette use. In contrast, 3.2% of the women reported daily snus use, 9.0% daily cigarette smoking and 2.4% any e-cigarette use. Among the tobacco use categories, daily snus users had the highest levels of cotinine. Of reported non-tobacco users, 3.5% had cotinine levels above the cut-off, compared with 68.0% among both occasional cigarette smokers and snus users, 67.5% among all e-cigarette users and 94.7% and 97.8% among daily cigarette smokers and snus users, respectively. Agreement between self-reported tobacco use and urinary cotinine was classified as strong for daily use of cigarettes (κ=0.824) and snus (κ=0.861), while moderate to weak for occasional smoking (κ=0.618), occasional snus use (κ=0.573) and any e-cigarette use (κ=0.576).

**Conclusions** We found high validity of self-reported tobacco use in our study population, particularly for daily tobacco use. Further, we found that daily snus users were exposed to high levels of cotinine. Together with previous findings, our results indicate good validity of self-reported tobacco use among young adults.

## INTRODUCTION

Research on health effects of tobacco mainly relies on self-reported exposure data. This type of exposure assessment is subjective and may result in under-reporting or over-reporting.[1] Objective assessment of tobacco use can be

## STRENGTHS AND LIMITATIONS OF THIS STUDY

⇒ Several types of tobacco use were validated using biochemical verification.
⇒ This is the first study to validate snus use in a population of young adults composed of both men and women.
⇒ A large study sample including both young men and women was analysed.
⇒ The study was conducted within a well-characterised, population-based birth cohort.
⇒ A time lag existed between reporting tobacco use and biochemical verification.

performed using cotinine, a metabolite of nicotine, as a biomarker in urine, blood or saliva.[2] However, these methods are expensive and time-consuming, and are therefore not feasible for large populations. Instead, biomarker data can be used to validate self-reported data in smaller subpopulations, to estimate validity in larger cohorts.

Self-reported cigarette smoking has been validated in several studies with varying results in different populations, but with generally high validity.[1] However, for other types of tobacco and nicotine products, such as smokeless tobacco and electronic cigarettes (e-cigarettes), validation studies are scarce.[3] In Sweden, the smokeless tobacco type snus is used to a higher extent than cigarettes.[4] While cigarette smoking has decreased in the Swedish adult population over the last several decades, to 6% daily smokers in 2022, daily snus consumption rates have increased to a current 14%, with the largest increases being among younger populations.[4] Men have a notably higher consumption of snus compared with women in Sweden, however, women's consumption is increasing rapidly. Although snus shares some active compounds with cigarettes, much less is known about its

health effects, particularly among young adults.[5] Self-reported snus use has been validated among Swedish pregnant women, where high agreement between self-reported snus use and urinary cotinine levels was found.[6] Still, no study has investigated the validity of self-reported information on snus use among young men and non-pregnant women.

E-cigarette use has increased globally the past decade, especially among young populations. In Sweden, an estimated 2% of the population use e-cigarettes, with the highest prevalence among men under 30 years.[4 7] Although initially intended to reduce the harmful effects of cigarettes by facilitating smoking cessation, e-cigarettes are currently under debate as to whether they rather cause adverse health effects among adolescents and young adults.[8] Still, more research is needed to evaluate potential health effects of e-cigarettes.[9]

With the need for research on health effects of tobacco and nicotine products among young adults comes the need for reliable exposure data. Further, as validity of self-reported tobacco use may change over time and be influenced by factors such as age and gender, as well as smoking habits and attitudes towards tobacco products, it is important to regularly perform validation studies in different populations. Therefore, the aims of this study were to describe the prevalence of cigarette smoking, snus use and e-cigarette use as well as to validate self-reported use of these tobacco types using urinary cotinine among young adults in a Swedish cohort.

## METHODS
### Study population
This cross-sectional study was conducted within the population-based Swedish birth cohort BAMSE (Swedish abbreviation for Children, Allergy, Milieu, Stockholm, Epidemiology). Detailed information on the BAMSE study is provided elsewhere.[10 11] The cohort consists of 4089 newborn children recruited between 1994 and 1996 in Stockholm, Sweden. The participants have been followed up with repeated questionnaires on lifestyle and health up to approximately 24 years of age. On several occasions, participants who answered the questionnaire were invited to clinical examinations. At the follow-up at around age 24, hereafter called the 24-year follow-up, conducted between December 2016 and May 2019, the response rate of the questionnaire and the attendance rate at the clinical examination were 75% (n=3 064) and 56% (n=2 270) of the original cohort, respectively.

### Patient and public involvement
Participants or the public were not involved in the design, conduct, reporting, or dissemination plans of the study.

### Self-reported tobacco use
During the 24-year follow-up, participants provided self-reported information on tobacco consumption in a web questionnaire. The current study included all participants who provided information on use of cigarettes, snus and e-cigarettes (n=3052), hereafter referred to as the study population. To capture long-term patterns and habitual tobacco use, participants were asked 'do you smoke (cigarettes)?' (answer options: 'No'; 'No, but I used to smoke'; 'Yes, sometimes'; 'Yes, daily'), 'do you use snus?' ('No'; 'No, but I used to use snus'; 'Yes, sometimes'; 'Yes, daily'), 'do you use electronic cigarettes?' ('No'; 'Yes, sometimes'; 'Yes, daily') and 'do you smoke waterpipe?' ('No'; 'Yes, every month'; 'Yes, every week'; 'Yes, daily'). For cigarettes and snus, the participants were asked to estimate average consumption per day (for daily smokers), week (for daily snus users and occasional cigarette smokers) or month (for occasional snus users) using open-ended questions. Age of regular use debut (at least once per week) of snus or cigarettes was also assessed. In addition, participants were asked whether they had used nicotine replacement products on the day of urine sampling.

### Validation population
To validate self-reported tobacco use, a subsample of the BAMSE 24-year follow-up was chosen for urine analysis of cotinine. The selection was conducted among participants who answered the questionnaire and donated a urine sample during the clinical examination at the age of 24 years. By design, all who reported daily smoking, daily or occasional snus use, daily or occasional e-cigarette use as well as all participants who provided a urine sample at a previous follow-up at the age of 4 years were included. The selection criteria resulted in 429 men and 439 women. An additional random selection of participants was conducted to reach a subsample of 500 men and 500 women. One participant with missing information on cigarette use and one participant who reported use of nicotine replacement products on the day of urine sampling were excluded, resulting in a validation population size of n=998. See online supplemental figure S1 for an overview of the selection process.

### Cotinine analysis
Urine samples collected at the clinical examination during the 24-year follow-up were analysed for cotinine at the Division for Occupational and Environmental Medicine at Lund University. Samples were stored in −80°C until analysis. Urinary cotinine was analysed using a modified method for serum[12] using liquid chromatography tandem mass spectrometry (LC-MS/MS; QTRAP 5500, Framingham, Massachusetts, USA). Briefly, urine samples were prepared in 96-well plates, diluted in buffer and hydrolysed using β-glucuronidase after deuterium-labelled internal standard was added. Samples were analysed randomised, and each batch was analysed with calibration standards, quality controls and chemical blank samples (Milli-Q water). The limit of detection (LOD) for cotinine was set to 1 ng/mL. Specific gravity (SG) was used to compensate for urine sample dilution and was measured using a digital refractometer. The analysis of cotinine is part of a quality control programme between

analytical laboratories coordinated by the University of Erlangen-Nuremberg, Germany.

## Statistical methods

Comparisons of background characteristics and tobacco use between men and women were performed with two-tailed t-tests, Wilcoxon rank-sum test or Pearson's $\chi^2$ test, as appropriate. The average consumption of cigarettes was categorised as <10, ≥10 cigarettes per day for daily smokers, based on an a priori decided cut-off. Snus consumption was categorised as <4 and ≥4 boxes of snus per week for daily snus users, based on sample distribution. Cotinine levels were adjusted for SG according to: $PSG=P*[(1.014)/(SG–1)]$, where $PSG$ is the adjusted cotinine concentration, $P$ is the detected cotinine concentration (ng/mL), 1.014 is the average SG in the validation population and $SG$ is the specific gravity of the individual urine sample.[13] Cotinine levels between tobacco types and gender were compared using Wilcoxon rank-sum test.

To assess the agreement between self-reported tobacco use and urinary cotinine levels, Cohen's Kappa coefficient was calculated at a cut-off level of 50 ng/mL cotinine, as recommended by a subcommittee of the society for tobacco and nicotine research.[14] Kappa coefficients were interpreted according to McHugh[15] as 0.00–0.20: no agreement, 0.21–0.39: minimal agreement, 0.40–0.59: weak agreement, 0.60–0.79: moderate agreement, 0.80–0.90: strong agreement and >0.90: almost perfect agreement. Sensitivity, specificity, positive predictive value (PPV) and negative predictive value (NPV) were also estimated. Analyses were stratified by type of tobacco (cigarettes, snus or e-cigarettes), daily or occasional use and gender. The use of waterpipe was not validated due to few users, but users of waterpipe were considered as tobacco users. Sensitivity analyses were performed excluding participants with mixed tobacco use (ie, using more than one tobacco type) or with missing data on waterpipe use (n=117). Another set of analysis was performed excluding those who provided the urine sample more than 6 weeks after answering the web questionnaire (n=492).

A prediction model was performed using logistic regression to assess factors associated with false negative reports of tobacco use, so-called under-reporting (ie, reporting no tobacco use but having a urine sample with ≥50 ng/mL cotinine). Factors included in the final model were identified using a stepwise backward selection approach according to Kirkwood and Sterne,[16] eliminating factors at a threshold level of p<0.2 from a log-likelihood ratio test. Factors considered were identified from the literature and included gender, parental socioeconomic status and parental smoking obtained from the baseline questionnaire; educational level, occupation, asthma, former smoking, former snus use, secondhand smoke exposure, body mass index, weekday of urine sampling and time between web questionnaire and urine sampling obtained from the 24-year follow-up. For details on covariate definitions, see online supplemental file 1. Participants with missing data were excluded from the stepwise backwards selection process, in total affecting 4.8% of participants. The highest rate of missing data was found for second-hand smoke exposure (2.4%), followed by parental socio-economic status (1.6%).

Statistical analyses were performed in STATA (V.16; Stata Corp). Statistical significance was considered at p<0.05.

## RESULTS

### Description of study population

An overview of sociodemographic characteristics of the study population can be seen in table 1. The mean age was 22.5 years (range 21.5–25.2 years), 53.0% were women and most participants were students (51.2%), followed by employed workers (40.9%) and other occupations (7.9%). The median time between answering the questionnaire and the urine sampling was 41 days (IQR 16–84 days).

### Self-reported tobacco use

Self-reported tobacco use is presented in table 2. Smoking was reported by 20.7% (7.5% daily and 13.3% occasionally), snus use by 15.6% (11.1% daily

**Table 1** Sociodemographic characteristics of the study population and the validation population at the BAMSE 24-year follow-up

| | Study population, n=3052 | Validation population, n=998 |
|---|---|---|
| | n (%) | n (%) |
| Male | 1435 (47.0) | 499 (50.0) |
| White collar at baseline | 2537 (84.5) | 811 (82.6) |
| Any parent smoking at baseline | 618 (20.4) | 209 (21.1) |
| Asthma | 345 (11.3) | 137 (13.7) |
| University degree at age 24 | 1093 (36.0) | 360 (36.2) |
| Main occupation | | |
| Student | 1561 (51.2) | 486 (48.7) |
| Working | 1246 (40.9) | 438 (43.9) |
| Other | 240 (7.9) | 74 (7.4) |
| | Mean (SD) | Mean (SD) |
| Age, years | 22.5 (0.6) | 22.4 (0.4) |
| BMI*, kg/m$^2$ | 23.1 (3.9) | 23.4 (4.1) |
| | Median (IQR) | Median (IQR) |
| Time between questionnaire and urine sampling* | 41 (16–84) | 42 (16–78) |

Percentages may not add up due to missing values.
*Data available from participants who attended the clinical examination at the 24-year follow-up, 74% (n=2264) of study population.
BMI, body mass index.

**Table 2** Self-reported tobacco use in the BAMSE cohort, at the 24-year follow-up

| | All, n=3052 n (%) | Women, n=1617 | Men, n=1435 | P value* (women vs men) |
|---|---|---|---|---|
| **Cigarettes** | | | | |
| Never | 2037 (66.7) | 1056 (65.3) | 981 (68.4) | 0.074 |
| Former | 382 (12.5) | 196 (12.1) | 186 (13.0) | 0.484 |
| Occasional | 405 (13.3) | 220 (13.6) | 185 (12.9) | 0.562 |
| Daily | 228 (7.5) | 145 (9.0) | 83 (5.8) | 0.001 |
| <10 cigarettes/day | *122 (54.2)* | *84 (58.3)* | *38 (46.9)* | *0.099* |
| ≥10 cigarettes/day | *103 (45.2)* | *60 (41.4)* | *43 (51.8)* | |
| **Snus** | | | | |
| Never | 2459 (80.6) | 1481 (91.6) | 978 (68.2) | <0.0001 |
| Former | 116 (3.8) | 27 (1.7) | 89 (6.2) | <0.0001 |
| Occasional | 138 (4.5) | 57 (3.5) | 81 (5.6) | 0.005 |
| Daily | 339 (11.1) | 52 (3.2) | 287 (20.0) | <0.0001 |
| <4 boxes of snus/week | *184 (54.4)* | *34 (65.4)* | *150 (52.4)* | *0.085* |
| ≥4 boxes of snus/week | *154 (45.6)* | *18 (34.6)* | *136 (47.6)* | |
| **E-cigarettes** | | | | |
| No | 2932 (96.1) | 1578 (97.6) | 1354 (94.4) | <0.0001 |
| Occasional | 105 (3.4) | 35 (2.2) | 70 (4.9) | <0.0001 |
| Daily | 15 (0.5) | 4 (0.3) | 11 (0.8) | 0.041 |
| **Waterpipe** | | | | |
| No | 2835 (97.9) | 1533 (97.9) | 1302 (97.8) | 0.997 |
| Occasional | 60 (2.1) | 32 (2.0) | 28 (2.1) | 0.895 |
| Daily | 2 (0.1) | 1 (0.1) | 1 (0.1) | 0.910 |
| **Number of tobacco types used** | | | | |
| 1 | 830 (79.2) | 406 (85.7) | 424 (73.9) | <0.0001 |
| >1 | 218 (20.8) | 69 (14.3) | 150 (26.1) | |
| **Secondhand smoke exposure** | | | | |
| Yes† | 89 (3.1) | 49 (3.2) | 40 (3.1) | 0.852 |
| | **Mean (range)** | | | **P value** |
| Age when starting to smoke cigarettes regularly | 17.1 (12–22) | 16.9 (12–22) | 17.4 (12–22) | 0.013‡ |
| Mean nr cigarettes/week (occasional smokers) | 8.3 (1–150) | 7.7 (0–50) | 9.0 (0–150) | 0.856§ |
| Mean nr cigarettes/day (daily smokers) | 9.0 (1–30) | 8.6 (1–25) | 9.7 (1–30) | 0.085§ |
| Age when starting to snus regularly | 18.5 (13–24) | 20.2 (15–23) | 18.0 (13–24) | <0.0001‡ |
| Mean nr boxes snus/month (occasional snus users) | 2.5 (0–15) | 1.8 (0–10) | 3.0 (0–15) | 0.003§ |
| Mean nr boxes snus/week (daily snus users) | 3.6 (0.5–10) | 3.0 (0.5–7) | 3.7 (0.5–10) | 0.004§ |

*P value from Pearson's $\chi^2$ test.
†Defined as daily exposure to indoor tobacco smoke.
‡P value from two-tailed t-test.
§P value from Wilcoxon rank-sum test.

and 4.5% occasionally) and e-cigarettes by 3.9% (0.5% daily and 3.4% occasionally). Women reported daily cigarette smoking to a larger extent than men, while men reported higher snus and e-cigarette use. A total of 34.3% of the participants reported some type of tobacco use, and among these, 20.8% used more than one type. Among daily smokers, the average number of cigarettes per day was 8.6 among women and 9.7 among men (p=0.085). The average number of boxes per week among daily snus users was significantly higher among men than women (3.0 vs 1.8, p=0.003). Women began to smoke cigarettes regularly at a younger age than men, while men were younger when beginning to use snus regularly. Secondhand smoke exposure was reported

by 3.1% (5.7% among tobacco users and 1.9% among non-users).

## Description of validation population

The sociodemographic characteristics of the validation population did not differ from the total study population (table 1), apart from a slightly higher prevalence of men (50.0% vs 47.0%) and current asthma (13.7% vs 11.3%). As a consequence of the inclusion criteria, self-reported tobacco use in the validation population was higher than in the total study population, with 51.0% using any tobacco (daily or occasionally). A summary of tobacco use in the validation population is displayed in online supplemental table S1.

## Urine cotinine levels

Cotinine levels by tobacco type are shown in figure 1 and detailed urine cotinine levels are presented in online supplemental table S2. In total, 76.2% of the samples had cotinine levels above the LOD and 44.4% had levels over the 50 ng/mL cut-off for validation. For all types of tobacco, daily users had significantly higher median cotinine levels than occasional users; 2094 ng/mL vs 422 ng/mL (p<0.0001) for smokers, 3599 ng/mL vs 345 ng/mL (p<0.0001) for snus users and 2749 ng/mL vs 527 ng/mL (p=0.019) for e-cigarette users. This pattern remained after exclusion of participants with mixed tobacco use, although cotinine levels were generally lower in each group. Among mixed tobacco users, the median cotinine level was 3073 ng/mL. Significantly higher cotinine levels were observed among high compared with low consumption of daily cigarette smoking and snus use (≥10 cigarettes/day or ≥4 boxes/week) after excluding other tobacco use (online supplemental figures S2 and S3). Men had higher cotinine levels than women among occasional cigarette smokers (1595 ng/mL vs 159 ng/mL, p=0.0003) and daily snus users (3900 ng/mL vs 2762 ng/mL, p=0.006). However, no significant gender differences were observed after excluding mixed tobacco users. The median level for non-users was 1.6 ng/mL. Non-tobacco users exposed to daily secondhand smoke (n=7) had a median cotinine level of 2.47 ng/mL.

## Validation of self-reported tobacco use

Table 3 displays validation statistics of self-reported tobacco use against urinary cotinine levels, by category of tobacco use. The overall agreement was 86.5%, the overall false negative rate (ie, under-reporting) was 3.5% and the overall false positive rate (ie, over-reporting) was 10.0%. Among the under-reporters, 40.0% reported former tobacco use and among the over-reporters, 88.0% reported occasional tobacco use.

For both daily cigarette smoking and snus use, the agreement between self-reports and urinary cotinine was classified according to McHugh as strong (κ=0.824 and κ=0.861, respectively), with high sensitivity (80.2% and 83.6%, respectively) and specificity (98.3% and 99.1%, respectively). For occasional smoking and snus use,

agreement and sensitivity were lower at the chosen cut-off level (κ=0.618 and κ=0.573, respectively), while specificity remained high. For any use of cigarettes and snus (daily and occasional combined), the agreement, sensitivity and specificity were high, with moderate agreement for smoking (κ=0.762) and strong for snus (κ=0.814). For any e-cigarette use, the agreement was weak (κ=0.576), with lower sensitivity (60.7%) compared with any smoking (86.6%) and any snus use (86.8%), but with high specificity (94.6%). No separate analyses were made for daily or occasional use for e-cigarettes, due to few daily users. However, all daily users of e-cigarettes had cotinine levels above the 50 ng/mL cut-off. NPV was consistently high across tobacco categories, while PPV was high for daily tobacco use (94.7% for cigarettes and 97.8% for snus) and somewhat lower for occasional use (68.0% for cigarettes and snus) and any e-cigarette use (67.5%).

Stratified analyses by gender revealed higher validity among men for snus and e-cigarette use, while no major gender differences were observed for cigarette smoking (see online supplemental table S3 for details).

## Sensitivity analyses

When excluding mixed tobacco users, agreement was generally somewhat lower compared with the analyses with the full validation population (see online supplemental table S4 for details). In contrast, the agreement tended to be somewhat higher after exclusion of participants who provided urine samples ≥6 weeks after answering the questionnaire, however, the interpretation of Kappa values was generally not influenced (see online supplemental table S5 for details).

## Prediction model for under-reporting

Using the backward stepwise selection approach, the factors former smoking, former snus use and time between questionnaire and urine sampling were included in the final prediction model. Former snus users were more likely to under-report tobacco use (OR: 5.7, 95% CI: 2.2 to 15,0), and a non-significant association was observed for former smoking (OR: 2.1, 95% CI: 0.9 to 4.6). Providing the urine sample more than 6 weeks after the questionnaire, as compared with within 6 weeks after the questionnaire, was also predictive of under-reporting (OR: 1.5, 95% CI: 0.7 to 2.9), although not statistically significant. The performance of the prediction model, estimated by the area under receiver operating characteristic curve, was 0.66.

## DISCUSSION

Among young adults from a Swedish population-based cohort study, we observed high validity of self-reported tobacco use, particularly among daily users, while somewhat lower for occasional use as well as for e-cigarette use at the cut-off level of 50 ng/mL urinary cotinine. Specificity and NPV of the questionnaire were consistently high across tobacco type and frequency of use, while

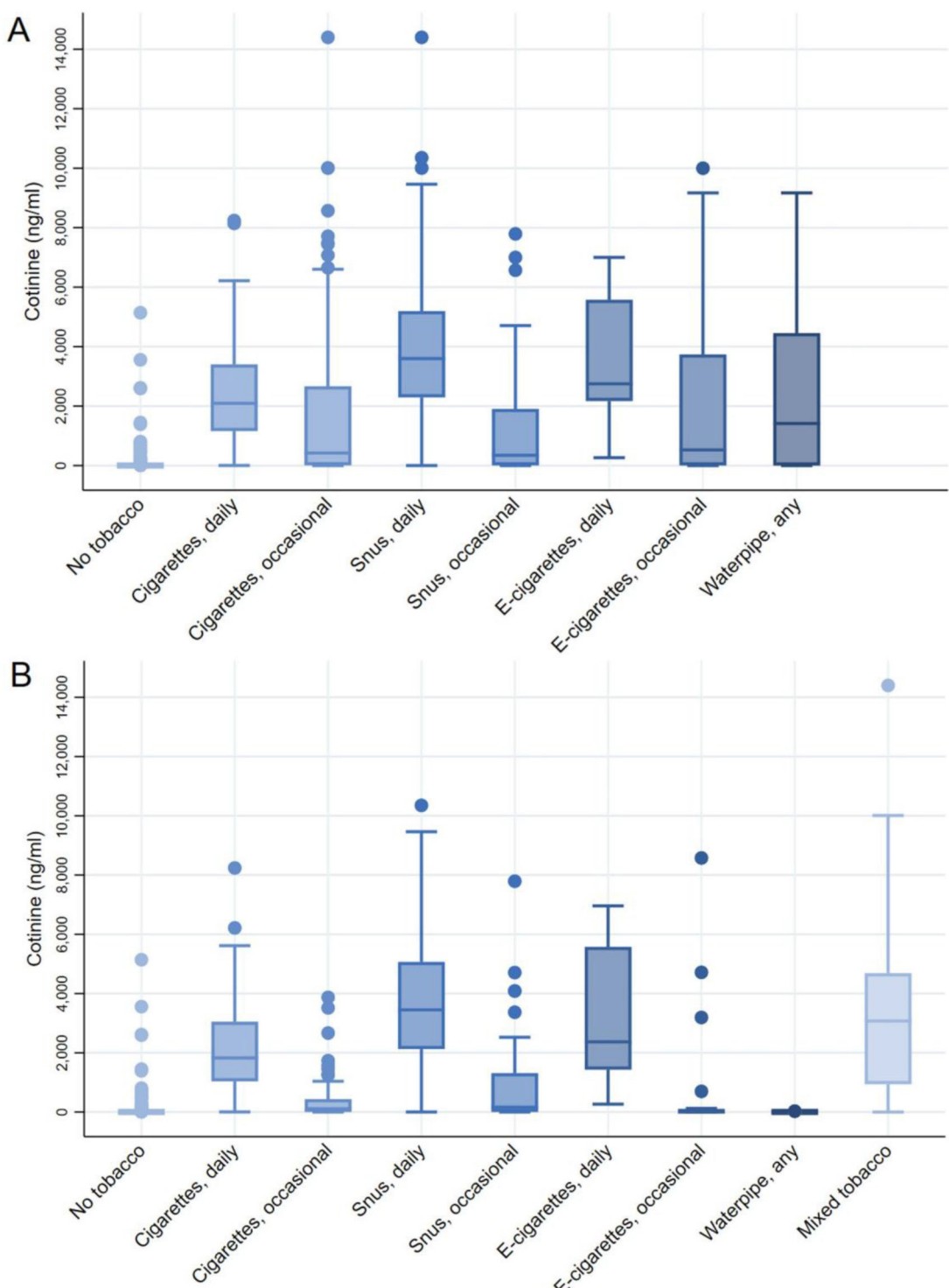

**Figure 1** (A) Density adjusted urinary cotinine levels by self-reported tobacco use. No tobacco (n=490); cigarettes, daily (n=150); cigarettes, occasional (n=125); snus, daily (n=182); snus, occasional (n=75); E-cigarettes, daily (n=9); E-cigarettes, occasional (n=71); waterpipe, any (n=26). Including participants with mixed tobacco use. (B) Density adjusted urinary cotinine levels among participants without mixed tobacco use as well as all mixed users. Cigarettes, daily (n=112); cigarettes, occasional (n=64); snus, daily (n=119); snus, occasional (n=55); E-cigarettes, daily (n=7); E-cigarettes, occasional (n=25); waterpipe, any (n=9); mixed tobacco (155).

sensitivity and PPV varied and was somewhat lower, especially for occasional tobacco use. Former use of tobacco and a longer time between answering the questionnaire and providing a urine sample were predictive of under-reporting, although former snus use was the only statistically significant prediction factor. Further, there were significant gender differences in tobacco use. Daily consumption of snus was high among men, while daily

**Table 3** Self-reported tobacco use against urinary cotinine levels by tobacco use

| Self-reports | Urinary cotinine level | | | | | | | |
|---|---|---|---|---|---|---|---|---|
| **Cigarette smoking** | | | | | | **Daily** | **Occasional** | **Any** |
| | <50 ng/mL | ≥50 ng/mL | *Total* | κ: | | 0.824 | 0.618 | 0.762 |
| No | 455 | 35 | *490* | Sensitivity: | | 80.2% | 70.8% | 86.6% |
| Occasional | 40 | 85 | *125* | Specificity: | | 98.3% | 91.9% | 90.5% |
| Daily | 8 | 142 | *150* | PPV: | | 94.7% | 68.0% | 82.5% |
| Total | *503* | *262* | *765* | NPV: | | 92.9% | 92.9% | 92.9% |
| **Snus use** | | | | | | **Daily** | **Occasional** | **Any** |
| | <50 ng/mL | ≥50 ng/mL | *Total* | κ: | | 0.861 | 0.573 | 0.814 |
| No | 455 | 35 | *490* | Sensitivity: | | 83.6% | 59.3% | 86.8% |
| Occasional | 24 | 51 | *75* | Specificity: | | 99.1% | 95.0% | 94.2% |
| Daily | 4 | 178 | *182* | PPV: | | 97.8% | 68.0% | 89.1% |
| Total | *483* | *264* | *747* | NPV: | | 92.9% | 92.9% | 92.9% |
| **E-cigarette use** | | | | | | **Any** | | |
| | <50 ng/mL | ≥50 ng/mL | *Total* | κ: | | 0.576 | | |
| No | 455 | 35 | *490* | Sensitivity: | | 60.7% | | |
| Occasional | 26 | 45 | *71* | Specificity: | | 94.6% | | |
| Daily | 0 | 9 | *9* | PPV: | | 67.5% | | |
| Total | *481* | *89* | *570* | NPV: | | 92.9% | | |

Cigarette smoking: analysis including all self-reported cigarette smokers and all non-tobacco users.
Snus use: analysis including all self-reported snus users and all non-tobacco users.
E-cigarette use: analysis including all self-reported e-cigarette users and all non-tobacco users.
Daily: results from analyses excluding occasional tobacco users.
Occasional: results from analyses excluding daily tobacco users.
Any: results from analyses combining daily and occasional tobacco users.
κ: Cohen's kappa coefficient.
NPV, negative predictive value; PPV, positive predictive value.

cigarette smoking was more common among women. Daily e-cigarette use was uncommon among both sexes. Daily snus users had the highest urine cotinine levels.

This study has several strengths, including a large and well-characterised study population from a population-based cohort, which provided power for the statistical analyses and allowed for stratification by tobacco type and gender. Furthermore, the study had repeatedly collected information on tobacco exposure, as well as sociodemographic factors. However, a limitation of the study design is that the questionnaire and the urine sampling were not conducted at the same occasion, and some participants may have changed their tobacco habits between the two assessments. The importance of time between the two assessments was apparent from both the sensitivity analysis as well as from the prediction model. Another limitation is the lack of information on specific tobacco brands or types, such as nicotine-free products, which may further cause misclassification of certain self-reported tobacco users. Similarly, use of nicotine replacement products may explain cotinine levels above the cut-off level among some of the former tobacco users.[17] However, only one participant in the validation populations reported using nicotine replacement products at

the day of urine sampling and was thus excluded from the analyses. Further, selection bias cannot be entirely ruled out as the validation population was selected from those attending the clinical examination, if non-attendance at clinical examination was related to accuracy in self-reporting.

Our study shows similar prevalence of tobacco use among young adults compared with data for the age group 16–29 years from Swedish national surveys conducted in 2016, 2018 and 2020.[18 19] From 2016–2020, daily smoking was reported by 4%–8% of young adults, daily snus use by 10%–11% and any e-cigarette use at 2%–3%, as compared with 7%, 11% and 4%, respectively, in our study. On the other hand, occasional smoking was somewhat higher in our study (13% vs 8%–11%). Further, the observed gender differences in our study were similar to those of a recent study among Swedish 19-year olds[20] and with figures from the Swedish national survey, describing higher snus and e-cigarette use among young men compared with women, but no difference in smoking prevalence between the genders.[18 19] However, a recent study on pooled data from two population-based Swedish cohort also found that cigarette smoking was more common among women across the age group 20–75 years.[7]

Urinary cotinine levels were consistent with self-reported frequency of tobacco use. Daily users had higher cotinine levels than occasional users, and a dose–response trend was observed for the amount of cigarettes and snus consumed. Daily snus users had the highest median cotinine levels, followed by e-cigarette and cigarette users. Smokeless tobacco users have previously been found to have higher urinary cotinine levels than cigarette smokers,[21 22] while the difference between cigarette smokers and e-cigarette users appears less consistent.[22–24] However, e-cigarette users were few in our study and the resulting cotinine levels may not be generalisable to other populations.

The high validity of self-reported tobacco use found in our study is consistent with previous findings. Several validation studies on self-reported cigarette smoking have found a high correlation between urinary cotinine and self-reported smoking status in population-based cohorts, including young adult populations.[25–30] In a cohort of young men and women in the USA (aged 18–30 years), the overall agreement between self-reported cigarette smoking (at least 5 cigarettes per week) and urinary cotinine at a cut-off level of 14 ng/mL was 95.8%.[30] Similarly, high sensitivity (93%) and specificity (99%) for self-reported smoking compared with a cotinine cut-off level of 25 ng/mL was found in a cohort of Malaysian male university students.[29]

In a previous validation study of self-reported snus use, high validity was found among Swedish pregnant women and adolescents (mean age 15 years).[6 31]

To date, few studies have validated self-reported e-cigarette use against biomarkers in population-based cohorts. Boykan et al[32] found a low (37%) concordance among adolescent self-reported e-cigarette users in the past week. The lower agreement observed for self-reported e-cigarette use in our study as compared with cigarettes and snus is likely a consequence of that most of the e-cigarette use was occasional, which was also true in the study by Boykan et al. In fact, in our study all daily e-cigarette users had cotinine levels above the cut-off value, although they were very few. Further studies with larger sample sizes are needed to assess validity of self-reported e-cigarette use more confidently. Some discrepancies between genders were observed for validity of self-reporting where men had generally higher validity, particularly for snus and e-cigarette use where the prevalence was higher among men. However, gender was not related to under-reporting. In studies conducted in the UAE, Georgia and South Korea, much lower validity for self-reported smoking was found among women compared with men, however, smoking rates were also much lower among women in these studies.[25 33 34] In contrast, in studies from Canada, the USA and Mexico, where smoking rates were more even between genders, similar validity was also observed.[28 30 35] These discrepancies may be related to cultural differences between the geographic regions of the studies.

We found low rates of under-reporting in our study population, in a similar or lower range than other study populations from the general population.[25–28 30 31 36] However, it should be noted that rates of under-reporting tend to be notably higher among certain patient groups, for example, chronic obstructive pulmonary disease patients,[37] head and neck cancer patients,[38 39] cardiovascular disease patients,[40 41] as well as among pregnant women,[6 42 43] especially when investigating smoking cessation. Although some nicotine exposure may occur from secondhand smoke exposure or diet,[44 45] such exposure is unlikely to cause misclassification at a cut-off level 50 ng/mL. Instead, the strongest predictive factor of under-reporting was former tobacco use. This is in line with previous findings,[28 30] and may be explained by relapse or use of other nicotine products during the study period. Some over-reporting was also found in the validation population, particularly among occasional tobacco users. Lower agreement in this group is expected compared with daily tobacco users at the same biochemical cut-off level, as their nicotine exposure would vary more over time. The same discrepancies between these groups and similar level of agreement between self-reported occasional cigarette smoking and serum cotinine were observed in a recent study from The National Health and Nutrition Examination Survey (NHANES).[36] As the half-life of cotinine is 12–20 hours,[2] it is likely that occasional users will have urinary cotinine levels similar to non-users only a few days after their latest use. Validity among occasional tobacco users is therefore likely not fully captured with a single urine sampling or at the chosen cut-off level.

As the validation was performed in a subsample of the BAMSE cohort, the results may not be entirely representative of the full cohort. Despite being enriched with tobacco users, the validation population was generally representative of the study population in sociodemographic factors, which indicates high validity in the whole cohort. However, as there were proportionally more non-tobacco users in the study population than the validation population, the rate of under-reporting may be somewhat higher in the study population. Similarly, the number of over-reporters in the study population may be lower, due to tobacco users being proportionally fewer. Still, the results of the validation analyses will be applicable to future research conducted on tobacco exposure in the BAMSE cohort. From a broader perspective, these results together with previous findings indicate that self-reported data on tobacco consumption can be used to accurately assess exposure for young adults. Questionnaires are important tools to study health effects of tobacco and can be used to meet the demand on more research on both e-cigarettes and snus. Although snus is banned in most EU countries[46] and is not a globally used type of tobacco, there has been a rise in snus sales in the USA over the past years.[47] This spread on the market increases the need to uncover adverse health effects of snus, in order to promote global public health.

## CONCLUSION

In this study, we observed significant gender differences in tobacco use among Swedish young adults, with a notably high prevalence of daily snus consumption among young men. Snus users had the highest levels of urinary cotinine. Further, we found high validity of self-reported tobacco use, particularly among daily tobacco users, with very low rates of under-reporting. These results indicate that self-reported tobacco use is valid for exposure assessment among young adults, which is important for future research on health effects of tobacco and nicotine in this age group.

**Author affiliations**
[1]Institute of Environmental Medicine, Karolinska Institutet, Stockholm, Sweden
[2]Department of Cardiology, Danderyd University Hospital, Stockholm, Sweden
[3]Division of Occupational and Environmental Medicine, Lund University, Lund, Sweden
[4]Centre for Occupational and Environmental Medicine, Region Stockholm, Stockholm, Sweden
[5]Sachs' Children and Youth Hospital, Södersjukhuset AB, Stockholm, Sweden
[6]Department of Clinical Science and Education, Södersjukhuset, Karolinska Institutet, Stockholm, Sweden

**Acknowledgements** We thank the children and parents participating in the BAMSE cohort and all staff involved in the study throughout the years. Moreover, we wish to thank Anna Rönnholm, Marie Bengtsson and Åsa Amilon at the Division of Occupational and Environmental Medicine at Lund University, Sweden, for performing the analyses of chemicals.

**Contributors** AZ took part in the study design, performed the formal analyses and interpretation of the results, created tables and figures, wrote the original draft and acts as guarantor of the work. SS contributed to revising the manuscript. LP contributed to revising the manuscript and provided funding. PL contributed to revising the manuscript and provided funding. GP contributed to revising the manuscript. NA provided data curation, assistance with the formal analysis and contributed to revising the manuscript. CL provided resources for and was responsible for the urine analysis and contributed to revising the manuscript. AG contributed to revising the manuscript. IK contributed to revising the manuscript. EM contributed to revising the manuscript and provided funding. SE took part in the study design, provided data curation and assistance with the formal analysis, contributed to the supervision of the project was a major contributor in revising the manuscript. AB had a major role in study design, methodology, project administration and supervision of the project, provided funding and resources and was a major contributor in revising the manuscript. All authors read and approved the final manuscript.

**Funding** This work was supported by the Swedish Research Council (VR, grant number 2018-02524); the Swedish Heart and Lung Foundation; the Swedish Research Council for Sustainable Development (FORMAS, grant number 2016-01646); the Swedish Research Council for Working Life and Social Welfare; the Swedish Asthma and Allergy Association Research Foundation; the Swedish Environmental Protection Agency (grant numbers NV-09284-13 and NV-00175-15); Region Stockholm (ALF project, and for cohort and database maintenance); the European Research Council (grant number 757919) and Karolinska Institutet.

**Competing interests** None declared.

**Patient and public involvement** Patients and/or the public were not involved in the design, or conduct, or reporting, or dissemination plans of this research.

**Patient consent for publication** Consent obtained directly from patient(s).

**Ethics approval** This study involves human participants and was approved by the Regional Ethics Committee at Karolinska Institutet in Stockholm, Sweden (Dnr 2016/1380-31/2) and conducted according to the Helsinki Declaration. All participants provided written informed consent.

**Provenance and peer review** Not commissioned; externally peer reviewed.

**Data availability statement** Data are available upon reasonable request. The datasets generated and/or analysed during the current study are not publicly available, due to the dataset containing sensitive personal data, but are available from the corresponding author on reasonable request.

**ORCID iD**
Anna Zettergren http://orcid.org/0000-0002-9145-9444

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
