## [Reviewer comments · BMJ Open]

ARTICLE DETAILS

TITLE (PROVISIONAL)	Assessing tobacco use in Swedish young adults from self-report and urinary cotinine: a validation study using the BAMSE birth cohort
AUTHORS	Zettergren, Anna; Sompa, Shanzina; Palmberg, Lena; Ljungman, Petter; Pershagen, Göran; Andersson, Niklas; Lindh, Christian; Georgelis, Antonios; Kull, Inger; Melen, Erik; Ekström, Sandra; Bergstrom, Anna

VERSION 1 – REVIEW

REVIEWER	Daniela Gutiérrez-Torres National Cancer Institute Division of Cancer Epidemiology and Genetics
REVIEW RETURNED	20-Mar-2023

GENERAL COMMENTS	In this manuscript, the authors analyzed the prevalence of tobacco use (cigarette, snus, and e-cigarette) among young adults in Sweden and the validity of self-report by calculating agreement with measured urinary cotinine concentrations. Results showed that daily consumption of snus was high among men while daily cigarette smoking was more prevalent among women. The agreement between self-reported of tobacco use and urinary cotinine (cut-point: 50ng/mL) was strong for daily cigarette smokers and daily users of snus ($\kappa=0.824$ and 0.861, respectively) and moderate for occasional cigarette smokers and occasional snus users. The manuscript is well written and contributes to the global surveillance of non-cigarette tobacco products, especially smokeless tobacco, and electronic cigarettes. There are minor changes that could be done to improve the clarity of the paper. I have provided some suggestions and edits for the authors to consider below. Specific comments: 1. Methods: • Self-reported tobacco use. What was the question used to classify participant's cigarette, snus, and e-cigarette use? Current use (last 30 days), or recent use (on the day of urine sample)? Adding the definition in the methods paragraph or in the "variable definitions" section in the supplementary material.• Validation population. Page 6, Line 36. Edit the follow-up time from 4-year to 24-year. all participants who provided a urine sample at the 4-year follow-up.• Statistical methods. Page 7, Line 55. Consider changing the word "investigated" to "estimated". Sensitivity, specificity, positive predictive value (PPV) and negative predictive value (NPV) were also estimated.• Statistical methods. Page 8, Line 29. Consider adding the number
--

	of participants with missing data that were excluded from the analyses. 2. Results.  • Page 8, Line 42. Consider changing “background” for sociodemographic. An overview of sociodemographic characteristics of the study population can be seen in Table 1. 3. Supplemental material.  • Figure S1. Correct typos in the following boxes “Did not attend clinical examination...” and “Participants who donate urine sample at 4-year follow-up”
--	--

REVIEWER	Hong Xue George Mason University
REVIEW RETURNED	18-May-2023

GENERAL COMMENTS	This study examined the validity of self-reported tobacco use among young adults in the Swedish birth cohort BAMSE. The topic is important itself. However, I have some concerns:  1. Generalizability of the results. The validation sample was selected from the participants who attended the clinical examination. Those who were willing to attend the clinical examination could be more likely to truthfully report tobacco use than those who were not. Such potential participant differences limit the validity and generalizability of the results from the study. This needs to be justified. 2. Whether the validation sample is representative of the study population, i.e. the BAMSE cohort, is questionable. According to Table 2, in the BAMSE cohort, the prevalence of self-reported daily cigarettes use, snus use, e-cigarette use was 7.5%, 11.1%, and 0.5% respectively. According to Table S1, the prevalence of self-reported daily cigarettes use, snus use, e-cigarette use in the validation sample was 15%, 18.2%, and 0.9% respectively. The differences are substantial. The agreement/validity metrics obtained from the validation population may not be readily applicable to the study population. 3. For the logistic regression based prediction model, model prediction performance metrics should be reported, such as F-score, AUC.
--

VERSION 1 – AUTHOR RESPONSE

Reviewer: 1

Dr. Daniela Gutiérrez-Torres, National Cancer Institute Division of Cancer Epidemiology and Genetics
Comments to the Author:

In this manuscript, the authors analyzed the prevalence of tobacco use (cigarette, snus, and e-cigarette) among young adults in Sweden and the validity of self-report by calculating agreement with measured urinary cotinine concentrations. Results showed that daily consumption of snus was high among men while daily cigarette smoking was more prevalent among women. The agreement between self-reported of tobacco use and urinary cotinine (cut-point: 50ng/mL) was strong for daily cigarette smokers and daily users of snus ($\kappa=0.824$ and 0.861 , respectively) and moderate for occasional cigarette smokers and occasional snus users.

The manuscript is well written and contributes to the global surveillance of non-cigarette tobacco products, especially smokeless tobacco, and electronic cigarettes.

There are minor changes that could be done to improve the clarity of the paper. I have provided some suggestions and edits for the authors to consider below.

Answer: We would like to thank the reviewer for the positive feedback and valuable comments.

Specific comments:

1. Methods:

• Self-reported tobacco use. What was the question used to classify participant's cigarette, snus, and e-cigarette use? Current use (last 30 days), or recent use (on the day of urine sample)? Adding the definition in the methods paragraph or in the "variable definitions" section in the supplementary material.

Answer: We thank the reviewer for this important question. When assessing tobacco use in the questionnaire from the 24-year follow-up, the questions were designed to capture patterns of long term and habitual tobacco use, although the time frame was not specified. The questions were phrased similarly to those in the National public health survey conducted by the Public Health Agency of Sweden¹. The BAMSE 24-year follow-up questionnaire included the following questions and optional answers to classify the participants use of cigarettes, snus and e-cigarettes:

Do you smoke?

- No
- No, but I used to smoke
- Yes, sometimes
- Yes, every day

Do you use snus?

- No
- No, but I used to use snus
- Yes, sometimes
- Yes, every day

Do you use electronic cigarettes?

- No
- Yes, sometimes
- Yes, every day

Do you smoke waterpipe?

- No
- Yes, every month
- Yes, every week
- Yes, every day

The questions and answer options are reported in the method's section under Self-reported tobacco use, on page 6, lines 110-115.

The text has now been updated to clarify the formulations of the questions. It now reads:

"To capture long-term patterns and habitual tobacco use, participants were asked "do you smoke [cigarettes]?" (answer options: "No"; "No, but I used to smoke"; "Yes, sometimes"; "Yes, daily"), "do you use snus?" ("No"; "No, but I used to use snus"; "Yes, sometimes"; "Yes, daily"), "do you use electronic cigarettes?" ("No"; "Yes, sometimes"; "Yes, daily") and "do you smoke waterpipe?" ("No"; "Yes, every month"; "Yes, every week"; "Yes, daily").

1 Public Health Reporting - The Public Health Agency of Sweden. Accessed 23-06-07
<https://www.folkhalsomyndigheten.se/the-public-health-agency-of-sweden/public-health-reporting/>

• Validation population. Page 6, Line 36. Edit the follow-up time from 4-year to 24-year. all participants who provided a urine sample at the 4-year follow-up.

Answer: The text is correct where it says 4-year follow-up. The subsample selected for urine analyses included those who had provided a urine sample at a previous follow-up at age 4 years. This, in order to have the possibility to perform longitudinal analyses on urine data within BAMSE. The selection of the validation population was designed keeping the more overarching objective of the BAMSE study in mind.

In order to more clearly describe the selection process and distinguish between the follow-ups, the text has been changed to the following (page 6, lines 123-129):

“The selection was conducted among participants who answered the questionnaire and donated a urine sample during the clinical examination at age 24 years. By design, all who reported daily smoking, daily or occasional snus use, daily or occasional e-cigarette use as well as all participants who provided a urine sample at a previous follow-up at age 4 years were included.”

• Statistical methods. Page 7, Line 55. Consider changing the word “investigated” to “estimated”. Sensitivity, specificity, positive predictive value (PPV) and negative predictive value (NPV) were also estimated.

Answer: We thank the reviewer for this suggestion. The text has now been changed according to the suggestion (page 8 line 161).

• Statistical methods. Page 8, Line 29. Consider adding the number of participants with missing data that were excluded from the analyses.

Answer: We thank the reviewer for this suggestion. A total of 48 participants (4.8%) were excluded from the prediction model due to missing data. These participants were excluded in the stepwise backward selection process but were all included in the final prediction model in the end, since there was no missing among the factors included in the final model.

The following section has been edited to accommodate your suggestion (page 8 lines 176-179):

“Participants with missing data were excluded from the stepwise backwards selection process, in total affecting 4.8% of participants. The highest rate of missing data was found for second-hand smoke exposure (2.4%), followed by parental socioeconomic status (1.6%).”

2. Results.

• Page 8, Line 42. Consider changing “background” for sociodemographic. An overview of sociodemographic characteristics of the study population can be seen in Table 1.

Answer: We thank the reviewer for this suggestion. The text has now been changed accordingly (page 8, line 184).

3. Supplemental material.

• Figure S1. Correct typos in the following boxes “Did not attend clinical examination...” and “Participants who donate urine sample at 4-year follow-up”

Answer: We thank the reviewer for catching this typo. The typo from the first example has been corrected (see figure s1 in updated supplementary document). The second example is not a typo however, it is correct that it refers to the follow-up from when the participants were 4 years old. A clarification to the selection process has been added to the method section (see answer to previous comment under section 1: Methods, comment 2.)

Reviewer: 2

Dr. Hong Xue, George Mason University

Comments to the Author:

This study examined the validity of self-reported tobacco use among young adults in the Swedish birth cohort BAMSE. The topic is important itself. However, I have some concerns:

Answer: We would like to thank reviewer 2 for the valuable questions and suggestions.

1. Generalizability of the results. The validation sample was selected from the participants who attended the clinical examination. Those who were willing to attend the clinical examination could be more likely to truthfully report tobacco use than those who were not. Such potential participant differences limit the validity and generalizability of the results from the study. This needs to be justified.

Answer: We would like to thank the reviewer for this important comment. It is correct that there may be discrepancies between attendees and non-attendees of the clinical examination related to the accuracy of reporting tobacco use. Among those who answered the questionnaire at age 24 years, a higher proportion of females attended the clinical examination (78% among females vs. 70% among males). Gender may be related to the accuracy of reporting tobacco use. Our stratified analyses show no gender differences for validity of cigarette smoking, while we observed higher overall validity among men for snus and e-cigarette use. However, there were no differences in underreporting, specificity or NPV of tobacco use among males and females, and gender/sex was not a predictive factor of underreporting. Despite this, it can not be excluded that other factors potentially related to attending the clinical examination may also influence the accuracy of reporting tobacco use. The following sentence has been added to the discussion section (page 15-16, lines 309-311): "Further, selection bias cannot be ruled out as the validation population was selected from those attending the clinical examination, if non-attendance at the clinical examination was related to accuracy in self-reporting."

2. Whether the validation sample is representative of the study population, i.e. the BAMSE cohort, is questionable. According to Table 2, in the BAMSE cohort, the prevalence of self-reported daily cigarettes use, snus use, e-cigarette use was 7.5%, 11.1%, and 0.5% respectively. According to Table S1, the prevalence of self-reported daily cigarettes use, snus use, e-cigarette use in the validation sample was 15%, 18.2%, and 0.9% respectively. The differences are substantial. The agreement/validity metrics obtained from the validation population may not be readily applicable to the study population.

Answer: We thank the reviewer for bringing up this important issue. It is true that the validation sample differs from the total study population in terms of tobacco use. This is a consequence of the selection process for the urine analysis, which favored tobacco users in order to have a sufficient sample for the validation. This may result in decreased representativity of the results of the full BAMSE cohort. The issue is expanded on in the discussion section, where we explore what this difference would mean for the occurrence of overreporting and underreporting in the full cohort. Further, we examined how well the two groups corresponded with regard to other sociodemographic and other characteristics and found that the validation sample represented the BAMSE cohort quite well. There may still be factors that could not be accounted for that affect the validity of self-reports, however.

The following section in the Discussion has been changed to account for this issue (page 18, lines 371-379):

“As the validation was performed in a subsample of the BAMSE cohort, the results may not be entirely representative of the full cohort. Despite being enriched with tobacco users, the validation population was generally representative of the study population in sociodemographic factors, which indicates high validity in the whole cohort. However, as there were proportionally more non-tobacco users in the study population than the validation population, the rate of underreporting may be somewhat higher in the study population. Similarly, the number of overreporters in the study population may be lower, due to tobacco users being proportionally fewer.”

3. For the logistic regression based prediction model, model prediction performance metrics should be reported, such as F-score, AUC.

Answer: We thank the reviewer for this suggestion. The AUC for the prediction model is 0.66. This information has been added to the manuscript in the result section (page 15, line 284-285) and now reads:

Performance of the prediction model performance, estimated by area under ROC curve (AUC), was 0.66.